# Functional diversity in human song

**Lucas Colares** [1] *, **Ray Balieiro Lopes-Neto**[2], **Alexandre Sampaio de Siqueira**[3], **Camila Ferreira Leão**[4], **Arianne Flexa de Castro**[4], **Bárbara Dunck**[4,5]

1 Programa de Pós-Graduação em Biodiversidade Animal, Laboratório de Ecologia Teórica e Aplicada, Universidade Federal de Santa Maria, Santa Maria–RS, Brazil, 2 Programa de Pós-Graduação em Botânica Tropical, Museu Paraense Emílio Goeldi, Belém–PA, Brazil, 3 Programa de Pós-Graduação em Ecologia e Recursos Naturais, Universidade Federal de São Carlos, São Carlos–SP, Brazil, 4 Programa de Pós-Graduação em Ecologia, Universidade Federal do Pará, Belém–PA, Brazil, 5 Instituto Socioambiental e dos Recursos Hídricos, Universidade Federal Rural da Amazônia, Belém–PA, Brazil

* lucasfcolares@gmail.com

**Data Availability Statement:** A reproductive workflow of all our analysis, including the data we used, is available at GitHub and can be accessed using the link https://github.com/lucas-colares/data-pop.

## Abstract

Functional diversity–i.e., the diversity of morphophysiological characteristics of species in a biological community–revolutionized ecology in recent decades, shifting the focus of the field from species to ecosystems. While its ecological applications are known, its adaptability to other disciplines, specifically music, is explored here. We retrieved fourteen characteristics of 12,944 songs by the top 100 artists of the 2010s decade on four streaming platforms. Then, we calculated the three main components of functional diversity–richness, evenness, and divergence–to each artist using probabilistic hypervolumes. Furthermore, we investigated to what extent functional diversity and the traits of an artist, its albums and songs has an effect on their popularity across streaming platforms such as Spotify. High functional richness, where an artist's songs differ greatly sonically, correlated with increased listens of up to 244,300,000. This would lead to estimated profit earnings exceeding $1,000,000 per richness gain. Danceable, highly-energetic, melodic, pop, and, notably, melancholic songs, albums, and artists are more listened to than their counterparts in streaming services. We captured how patterns in human song might reflects the social state of human societies in recent years and demonstrate the potential of applying functional diversity concepts and tools across scientific and economic domains, extending its relevance beyond ecology. By demonstrating applications of state-of-the-art functional diversity metrics using music as a case study, we intent to communicate the often-complex concepts of functional diversity using the familiar realm of music, which is an intrinsic trait of human cultures across the globe.

## Introduction

In the past three decades, the study of biodiversity has undergone a significant transformation, placing a central focus on the functional diversity of traits–the diversity in morphophysiological and behavioral characteristics of individuals–as a key element in understanding ecosystem processes [1]. Concepts related to functional diversity remain subjects of intense debate,

**Funding:** The National Council for Scientific and Technological Development (CNPq) and the Brazilian Federal Agency for Support and Evaluation of Graduate Education (CAPES) provided fellowships to LFC during the development of this paper. The publication of this article was supported by the Federal University of Pará (UFPa) (PROPESP-PAPQ 01/2023). Despite this support, the funders had no role in the study design, data collection and analysis, decision to publish, or preparation of the manuscript.

**Competing interests:** The authors have declared that no competing interests exist.

despite their undeniable importance in current literature and recent efforts to standardize them [2–6]. However, there is a general consensus that functional diversity comprises three primary facets.

Firstly, functional richness quantifies the extent of the functional space occupied by a particular community or species [4]. If a community or species exhibits numerous individuals with diverse morphophysiological and behavioral traits, it will have high functional richness [4]. Secondly, functional evenness evaluates the regularity of abundance distribution across the functional space. It indicates the likelihood of a random individual from a species or community possessing a specific trait value within that space [4, 5]. Thirdly, functional divergence measures how the abundance distribution within the functional space maximizes divergence in the traits [4, 5].

These concepts may be applied to assess the functional diversity of communities or species [5]. However, functional trait-based approaches should remain flexible enough to be applied in other scientific fields, such as genetics, evolution [2], as well as in fields that are seemingly distant from biology, such as art, literature, architecture, and music [7]. Wherever a study object can be distinguished by its characteristics, functional diversity can provide valuable insights into overall patterns within those characteristics.

Music is an inseparable aspect of the cultural development of human societies, with records of instruments dating back to the Paleolithic period [8]. Throughout history, music has served various purposes, including interpreting nature, facilitating communication, fulfilling celebrations and religious gatherings [9–11]. Technological advancements in the 20th and 21st centuries, such as radio, LPs, cassette tapes, music videos on television, CDs, and more recently, the internet and streaming services, have revolutionized public consumption patterns and transformed the way artists and labels create and distribute music [12]. In 2020, the music industry earned $21.6 billion globally, the sixth consecutive year of growth [13]. This surge in competition and revenue decentralization has directly influenced marketing strategies and music production, leading to the emergence of genre-bending music and the utilization of social media platforms for marketing [14, 15].

The formula for achieving a successful career as a musical artist has captivated data scientists, music experts, economists, and even everyday listeners for decades [16, 17]. Many factors contribute to an artist's success, including marketing strategies employed by labels and engagement with fandoms [17–19]. However, it is crucial to consider the inherent traits of the songs released by an artist, as they directly influence how a song sounds. Distinctive characteristics of songs, such as beats per minute (BPM), valence (i.e., the positivity of a song), danceability, key, scale, and others, can be incorporated into functional metrics to differentiate artists based on their musical identities [17, 20]. For instance, the song "Royals" by Lorde has a signature tempo of 85 beats per minute, while Lady Gaga's "Applause" features a tempo of 140 beats per minute, allowing for clear differentiation between the two songs. By incorporating this functional trait-based approach into music, we gain valuable insights into the unique attributes that shape an artist's musical identity and contribute to their overall success.

Here, we explore the application of the ecological concept of functional diversity and its measures to the realm of music by analyzing the sound characteristics of the top 100 most influential artists of the 2010s decade. We further investigated the effects of functional diversity and traits on the popularity of artists, its albums and songs on Spotify. To achieve this, we compiled data on traits and popularity for a comprehensive dataset comprising 12,944 songs from 787 albums by the top 100 artists listed in Billboard's decade-end chart for the 2010s [21] and popularity data for these songs from four platforms, that is YouTube (https://www.youtube.com/), Spotify (https://open.spotify.com/), Genius (https://genius.com/) and Last.fm (https://www.last.fm/). We extracted thirteen measures (i.e., traits) for each song, spanning key, mode,

time signature, duration, acousticness, danceability, energy, instrumentalness, liveness, valence, speechiness, loudness, beats per minute and musical style of the songs. Using this data, we successfully established a crossover between functional diversity and music, enlightening functional diversity concepts and highlighting the significance of functional richness in shaping artists' popularity on streaming platforms.

## Materials and methods

### Most influential artists

We compiled the characteristics and popularity data for the songs of the top 100 artists in the 2010s decade-end chart of Billboard [21]. The decade-end chart considers how well an artist performed across the Billboard charts during the 2010s (i.e., songs and albums in the charts), their amount of sold music (either physical or digital sales) and financial success in tours [21]. We compiled data from studio albums, soundtracks, extended plays (EP), mixtapes or collaborative albums available on Spotify, Deezer and YouTube platforms (at least two of these) released until December 2020 that had at least 80% of its songs never released before. This led to the exclusion of live and remix albums, such as "Lady Croissant", by Sia, and "The Remix", by Lady Gaga. These criteria also led to the exclusion of mixtapes that were not released in the streaming platforms investigated in this study, such as "Playtime is Over", the first project ever released by Nicki Minaj. In our final dataset, we were able to retrieved popularity data for 12,876 songs across Spotify, Genius, Last.fm and YouTube, functional trait data (excluding musical genres) for 11,090 songs, and musical genre data for 11,825 songs. After compiling this data, we had information on number of streams, musical genres and functional traits for a total of 12,944 unique songs. After matching the songs across all these three databases, we retained 10,444 songs that were presented in all three datasets for the subsequent analysis.

### Functional characterization

We assessed fourteen characteristics (i.e., functional traits) for each song of the 100 artists: (i) key, (ii) mode, (iii) time signature, (iv) duration (in milliseconds), (v) acousticness, (vi) danceability, (vii) energy, (viii) instrumentalness, (ix) liveness, (x) valence, (xi) speechiness, (xii) loudness, (xiii) beats per minute, and (xiv) musical genres. We choose these traits because they summarize a comprehensive array of different song types and characteristics.

Acousticness is a confidence measure, with values closer to 1 indicating a high confidence that the track is acoustic (e.g., "Truce", by Twenty One Pilots, which has a acousticness value of 0.99). Danceability ranges from 0 to 1 and describes the suitability of a track for dancing, taking into consideration its tempo, rhythm stability, beat strength and overall regularity (e.g., spoken interludes such as "!!!!!!!" by Billie Eillish score near 0 in danceability, while Pitbull's "Go Girl" scores 0.98). Energy represents the perceptive intensity and activity of a song and takes into account dynamic range, perceived loudness, timbre, onset rate, and general entropy, with energetic tracks sounding fast, loud and noisy (e.g., "Hold Back", by The Rolling Stones scores 0.99 in energy, while Paul McCartney's "Lament. Lamentoso" scores 0.002 in energy). Instrumentalness ranges from 0 to 1 and predicts whether a track has no vocals, with values closer to 1 indicating an instrumental song (e.g., Lady Gaga's "Chromatica III" scores 0.98 in instrumentalness; vocalizations are treated as instrumentals in this metric). Liveliness measures the presence of an audience in the track, with values closer to 1 indicating a high probability of a live audience in the recording (e.g., Maroon 5's "Not Coming Home" from "Songs About Jane" scores 0.98 in liveliness). Valence, ranging from 0 to 1, indicates how positive a track sounds (e.g., Pharrell Williams's "Happy" scores 0.96 in valence). This metric was initially developed by The Echo Nest, where music experts classified songs on a scale from most

to least positive, and a machine learning model used this information to predict valence in other tracks [22]. Speechiness ranges from 0 to 1 and detects the presence of spoken-word in a track (such as poetry or audio book; e.g., Cardi B's "Get Up 10" from her debut album "Invasion of Privacy", which scores 0.96 in speechiness). Loudness is a measure of the average decibels (dB) of a track.

Functional traits from iv to xiii were treated as quantitative continuous traits, while key, mode and time signature (i to iii) were treated as multichoice nominal traits. Categories of key range from 0 to 11, with each of these numbers corresponding to C, C#, D, D#, E, F, F#, G, G#, A, A#, and B, respectively, while categories of mode are major (i.e., 1) or minor (i.e., 0) and time signature categories are 0/4, 1/4, 2/4, 3/4, 4/4 and 5/4. Musical genre was treated as a multichoice nominal variable coded by binary columns, as a song can resemble multiple genres, and was retrieved from the track tags on Genius website (https://genius.com/). For the final calculations of functional diversity, we considered only musical genres that appeared in at least 500 of all 10,444 songs, these are Country, Pop, Pop-Rock, Rap, Rock and Trap. Since not all song tags in the Genius database indicated musical genre, we removed country/city names (i.e., Canada, UK, West Coast), and other words that did not indicate a musical genre (e.g., memes, beef, singer-songwriter; see S1 Table in S1 File for a list of the Genius tags we removed from the analysis). Traits from i to xiii were retrieved from the Spotify application protocol interface (API), which is a virtual connection between two or more computers, databases or software–in this case, a connection between the R software [23] in our computer and the official Spotify database (https://developer.spotify.com/documentation/web-api).

## Popularity measures

We retrieved popularity data for each song of the artists from Spotify, Last.fm, Genius and YouTube platforms, measured as the total number of times the song was viewed/played in the platform. Spotify and Last.fm popularity data were compiled from the API of each corporation, while the Genius and YouTube data was scraped from their website. The compilation of the traits and popularity song data was conducted from June 2021 to January 2022. All our data collection compiles with the terms and conditions from the Spotify and Last.fm web API services for non-commercial purposes [24, 25].

## Functional indexes

Using the functional and popularity song data, we decomposed the functional diversity of each artist in three components: richness, evenness and divergence. All three components of functional diversity were calculated using the probabilistic hypervolume method adopting a proportion of the probability density of 0.99 [5, 26]. This threshold makes sure that we include a major part of the estimated density function but still limit 0.01 of density to control for outliers in the calculations [26]. This probabilistic framework relies on probability density functions using a gaussian kernel density estimation and it reflects the unequal probabilities of different traits being retrieved in a given community (see [26] for mathematical details). This method works on the assumption that, although a variety of trait values or combinations are possible within a community, there are a few values that are more likely to occur than others given the uneven dominance of species within a community [26]. In other words, this probabilistic method estimates the size and shape of the hypervolume occupied by species traits and then calculates which regions have a higher density (and thus, higher probability) of traits based on species abundance. To calculate the indexes using the probabilistic hypervolume method, we used the sum of the number of streams (i.e., total popularity data) of artists across Spotify, YouTube, Genius and Last.fm in an analogy with abundance data in four different

communities (i.e., four different streaming platforms) and the first two axis of a Principal Component Analysis (PCA) as proxy representations of the functional traits of an artist's songs [27]. The PCA summarized the fourteen traits of each artist, ordinating all songs of the artists according to their similarities and the traits that best explain its distribution in a multidimensional space [27]. Therefore, similar values in the PCA axis for different artists indicate that they have a similar body of work. All traits were standardized prior to conducting the PCA.

After summarizing the trait data using the PCA, we conducted the calculation of the three functional diversity indexes. Functional richness is a measure of the volume of the hypervolumes occupied by the traits of an artist's songs (i.e., the "size" of an artist's body of work) in which trait probability is higher than zero [5]. High values of functional richness indicate artists with a broad body of work, with many different songs [5]. The functional evenness component represents the regularity in the distribution of the characteristics of the songs [5]. In other words, functional evenness here is a measure of the overlap between the actual functional space of an artist and an imaginary functional space in which all traits are evenly distributed across all songs [5]. Finally, higher values of functional divergence indicate a collection of songs highly different among each other [5]. More specifically, functional divergence is a measured of average distance between random points in the hypervolume from its centroid distribution [5]. Ultimately, if two artists have a similar value for functional richness, evenness or divergence, that means they have a similar size, regularity and difference in their trait space, respectively. However, it is worth noticing that similar values of richness, evenness and divergence do not mean that different artists have a similar body of work (i.e., similar styles of song), but rather similar configurations of trait space within their own domains. In this sense, a rap artist may have similar functional richness than a country artist, which means that the multidimensional space occupied by the traits of their songs has a similar size within their own musical domains.

## Data analysis

To investigate how functional diversity is associated with the number of streams on each music platforms, we built one linear model. In this model, the three functional indexes and their interactions were treated as the predictor variables, while the log10 of the total number of times the songs of the artists were listened on Spotify was treated as the response variable. We choose to represent the popularity of artists in a logarithmic scale because of the great dispersion of this data, which ranges from 1,054,314,241 to 33,440,789,574 streams. We also included year of career launch in this linear model as a predictor of number of streams. We choose to implement year of career launch as a predictor of functional diversity because most of the streaming platforms considered in this study were launched around 2006 and started to change the way that music was consumed in the world [28]. Therefore, artists that started their career prior to the launch of streaming platforms might have been in a disadvantage since they could inevitably be less listened in these platforms [28]. In the end, the structure of our global linear model was as it follows: *log10(artists' popularity) ~ Richness + Evenness + Divergence + Year of career launch + Richness:Year of career launch + Evenness:Year of career launch + Divergence:Year of career launch + Richness:Evenness + Richness:Divergence + Evenness:Divergence + Richness:Evenness:Divergence*. We conducted a stepwise model selection (in both backward and forward directions) to retain the model with the lowest Akaike Selection Criteria (AIC) [29].

Finally, to reveal which traits had an effect on the number of times that artists, their albums and songs were listened to in Spotify, we conducted three global linear models, in which the

song traits (i.e., key, mode, time signature, duration, acousticness, danceability, energy, instrumentalness, liveness, valence, speechiness, loudness, beats per minute, and musical genres) were considered as predictors of the log10 of the total number of times the artists, their albums, and songs were listened to in Spotify. Following the construction of this linear model, we conducted a stepwise model selection and retained only the models with the lowest AIC [29]. We repeated the same procedures to investigate which traits drive popularity of artists, albums and track by each of the seven major musical genres (i.e., Country, Pop, Pop/Rock, Rock, Rap, Trap and R&B). We were not able to investigate which traits rive the popularity of R&B, Trap and Pop/Rock artists due to the few replicates at the organizational level of artist. At the end, we conducted 21 linear models, three global models (i.e., without musical genres separations at three different levels of organization: artists, albums and tracks) plus 21 models for each of the seven musical genres at three levels of organization minus the three models we did not conduct for a lack of replicates (i.e., Trap, R&B and Pop/Rock models at the level of artists).

Prior to the linear models, we standardized all traits and checked for multicollinearity between them. This step is important to investigate if any two traits have a high correlation between each other and, if so, only one was retained (i.e., adopting a correlation threshold of |0.6|). This step led to the removal of "acousticness" and "loudness" traits given their high correlation with the "energy" trait. We also removed the "rap" category in the "musical genre" trait given its high correlation with the speechiness trait. You may check the correlation matrix of traits and which traits were retained in the final model prior to stepwise selection at S2-S4 Tables in S1 File. We choose to include only Spotify streams in our linear models because the number of streams in other platforms are not as consistent as Spotify, as several other versions of the same songs are also available for listening in other platforms. All analysis were conducted in the R software [23]. All graphical representation of the results were constructed using the ggplot2 [30], ggpubr [31], ggrepel [32], ggtext [33], ggh4x [34], ggpolypath [35], RColorBrewer [36], and the R.utils [37] packages. Functional diversity metrics were calculated using the TPD package [26], and data treatment and standardization were conducted using the reshape2 [38] and vegan packages [39].

## Results

When we plotted the top and bottom five most and least functionally rich, even, and divergent artists, we were able to evidence and visually demonstrate the theoretical concepts of the three facets of functional diversity in a two-axis functional space that summarized 27% of the variance trait data. The top five functionally richest artists were XXXTentacion (Fig 1A), Beyoncé (Fig 1D), Lady Gaga (Fig 1G), Billie Eilish (Fig 1J) and Paul McCartney (Fig 1M), respectively. All of which presented a broader distribution in their functional space than the bottom five least rich artists (i.e., Jason Derulo, Fig 1P; Bon Jovi, Fig 1S; Flo Rida, Fig 1V; Thomas Rhett, Fig 1Y; and The Chainsmokers, Fig 1B). In terms of functional evenness, the top five consists of Kenny Chesney (Fig 1B), Blake Sheldon (Fig 1E), Lady A (Fig 1H), 5 Seconds of Summer (Fig 1K), and Eric Church (Fig 1N), respectively. The top five most even artists all presented a density distribution evenly spaced across the functional space in comparison to the bottom five least functionally even artists (i.e., Eminem, Fig 1Q; Lil' Wayne, Fig 1T; B.o.B., Fig 1W; Juice WRLD, Fig 1Z; and Meek Mill, Fig 1C). Finally, the top five most divergent artists were Eminem (Fig 1X), Lady Gaga (Fig 1F), Panic! At The Disco (Fig 1I), Calvin Harris (Fig 1I), and 5 Seconds of Summer (Fig 1O), all of which presented the densest part of trait distribution in areas far from the centroid distribution of trait space in comparison to the least divergent artists (Fig 1ruxAD).

As we tracked the number of times each song of the artists was played in Spotify and associated it with the components of functional diversity of those artists, we found that artists with a

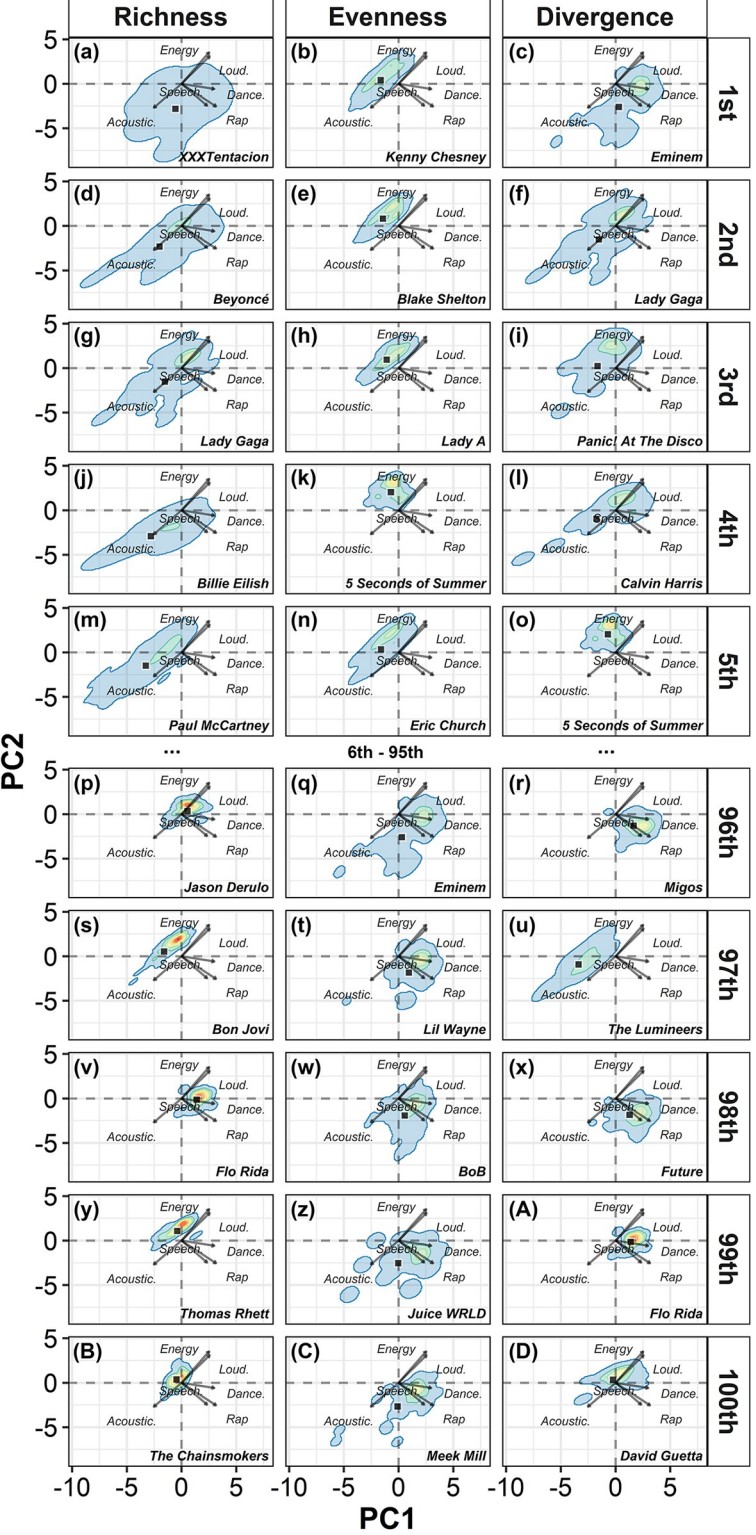

**Fig 1. Functional space of top and bottom artists.** Two-dimension functional space of the characteristics of the songs from the top and bottom five artists in terms of functional richness (panels in the left side), evenness (panels in the center), and divergence (panels in the right). The squares in the graph represents the centroid distribution of the traits of artists. Arrows in the center of the graph indicate which is the most defining characteristic of songs in that direction (i.e., only for traits in which loadings were higher than |0.3|). Variable loadings are provided in S5 Table in S1 File.

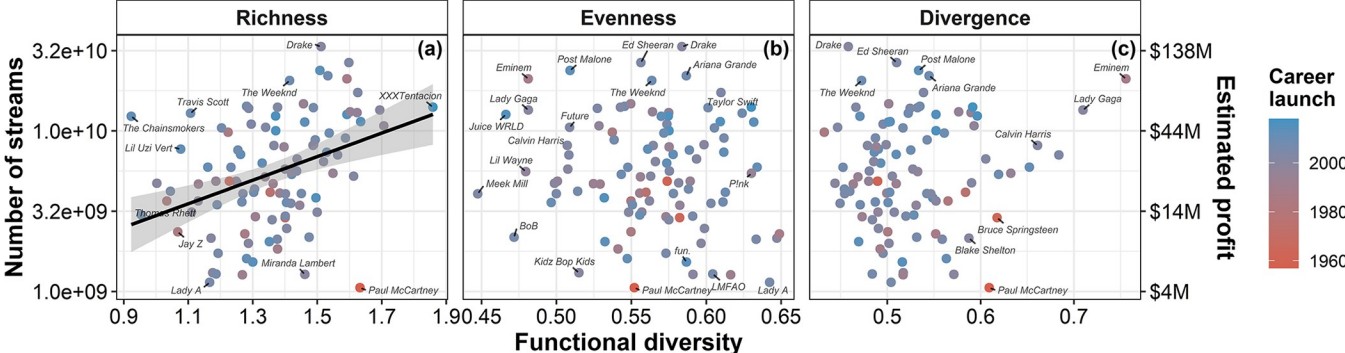

**Fig 2.** Associations between functional (a) richness, (b) evenness (c), and divergence of artists. Points are colored according to the year when artists launched their career. Second Y axis represents total profit considering that 1 stream on Spotify leads to $0.00437 of profit [40]. Total number of streams and profit are represented in a logarithm scale for better visualization of data distribution. Lines represent the direction of significant associations between the two variables and the shaded area represent its 95% confidence interval.

functionally rich catalog of songs are more listened to than artists with low functional richness, but this effect depends on when an artist has launched their career (Fig 2A; adjusted $r^2 = 0.18$; p<0.01). Interestingly, for every increase of 1 in an artist's functional richness, we observed a substantial increase of approximately 244,300,000 listens on Spotify alone. Functional evenness (Fig 2B) and divergence (Fig 2C) did not show any relevant influence on the popularity of artists (check S6 and S7 Tables in S1 File for full model coefficients and selection).

Overall, the songs released by the top 100 most influential artists of the 2010s are ~3 minutes and a half and 121 beats per minute, and are more danceable and highly energetic rather than instrumental, live and spoken. Furthermore, the songs we investigated in this study are fairly balanced between positive and depressed feeling and between keys, except for D sharp, which is exceptionally less used to build songs than other keys. The majority of the songs were pop, rap and rock songs (i.e., equivalent to ~75% of songs) built in a time signature of 4/4 (see S1 and S2 Figs in S1 File for the distribution of all individual traits).

We evidenced that the number of streams a song receives on Spotify is influenced by the traits of the artists (adj. $r^2 = 0.44$; p<0.01), albums (adj. $r^2 = 0.31$; p<0.01) and songs (adj. $r^2 = 0.23$; p<0.01; S8 and S9 Tables in S1 File). Danceability always had a positive influence on the popularity of artists, albums and songs (Fig 3). Whereas the positiveness of a song (i.e., valence), the amount of speech there is in a song and the rock genre always had a negative influence in the number of times an artist, albums and song were played in Spotify (Fig 3). When we investigated the traits that drive popularity within each musical genre, the results remained fairly similar to the global models (S3 Fig in S1 File). However, we observed contrasting patterns on the effect of speechiness and energy on popularity across different musical genres (S3 Fig, S10 and S11 Tables in S1 File). Although speechiness has a negative effect on popularity for most musical genres, it exerts a positive influence on the popularity of country artists and albums (S3 Fig in S1 File; p<0.01). Energy has a positive effect on the popularity of most musical genres but Rap and Trap, genres in which energetic music leads to low popularity (S3 Fig in S1 File; p<0.01).

## Discussion

### Functional diversity concepts through music

In ecology, high functional richness refers to a species or community with a wide distribution of traits in the functional space, either at one dimension, when only one trait is considered

(S4a Fig in S1 File), or in the multidimensional space (Fig 1). Otherwise, low functional richness occurs when the traits displayed by a species or community are narrowly distributed across the functional space (Figs 1 and S4a in S1 File). A community with high functional richness indicates that almost all the available resources and conditions in the habitat are utilized by some species or individuals [3, 4]. This can enhance ecosystem resilience by minimizing the potential negative outcomes of ecological invasions [40] and environmental fluctuations [4, 40]. To illustrate this concept in music, artists such as Beyoncé (Fig 1D), Lady Gaga (Fig 1G), Billie Eilish (Fig 1J), and Paul McCartney (Fig 1M) exhibited high functional richness, showcasing a diverse range of traits and encompassing various musical styles. Throughout their careers, these artists have released highly energetic, loud, and danceable songs, as well as acoustic tracks, exploring the musical spectrum between rock, pop and even jazzy sounds (Fig 1). In contrast, artists with low values of functional richness consistently produce similar types of songs throughout their careers (Figs 1 and S4a in S1 File).

We can interpret evenness as how evenly abundance is distributed within the functional space. When a biological community exhibits high functional evenness, it indicates that the resources and conditions are evenly exploited by all individuals of species [3, 4]. At the species level, high functional evenness implies that all individuals within a species equally exploit the entire functional space they occupy [5]. In our study, we represented abundance using popularity data, consequently, artists with high functional evenness (e.g., Kenny Chesney, Blake Shelton, Lady A, 5 Seconds of Summer, and Eric Church) are those whose all types of songs received a relatively similar level of attention (Figs 1 and S4b in S1 File). Whereas artists with low functional evenness (e.g., Eminem, Lil' Wayne, BoB, Juice WRLD, and Meek Mill) are characterized by an imbalanced distribution of listens across their songs within the functional space they occupy (Figs 1 and S4b in S1 File). This means that only a small portion of their songs received regular listens, while other types of songs are less listened to (Figs 2 and S4b in S1 File).

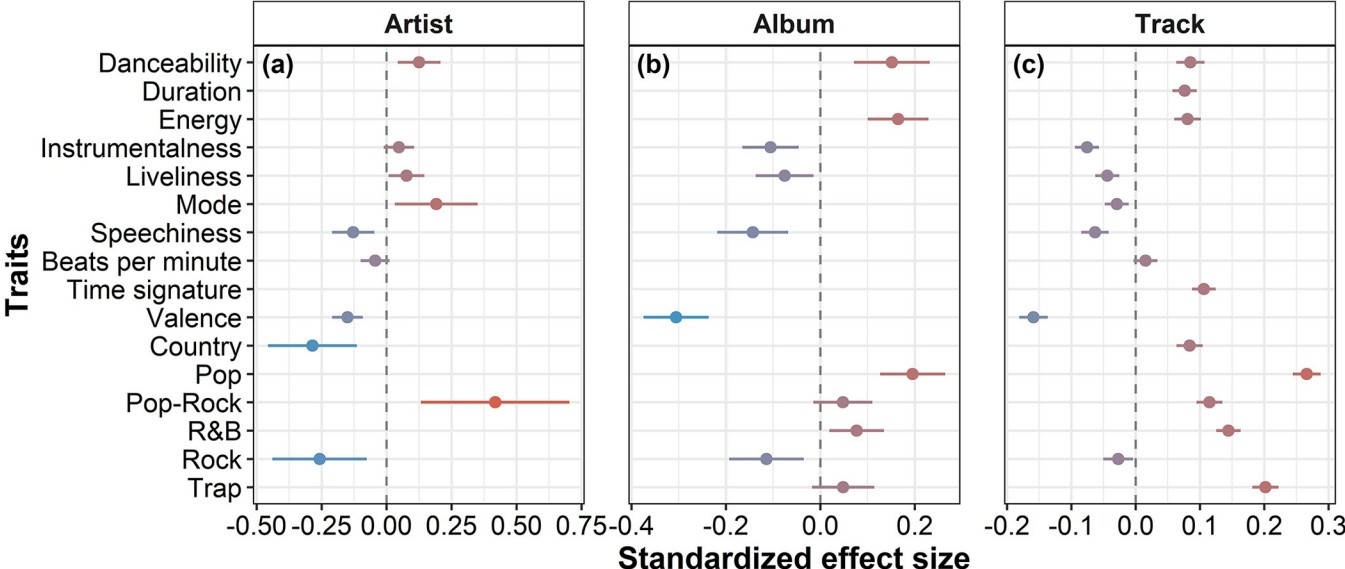

**Fig 3. Associations between popularity and traits.** Standardized effect size of the association between the selected traits and the number of times that (a) artists, (b) their albums, and (c) songs were played in Spotify. Whiskers represent the 95% confidence interval of the mean estimate, which is represented by the points. Color of whiskers represents the size of the effect in the negative direction (blue whiskers) and the positive direction (red whiskers). Traits that do not have a whisker in a plot were not selected in the linear model after stepwise regression (selected models in S8 and S9 Tables in S1 File).

Finally, the divergence component of functional diversity pertains to the trait distribution in relation to the center of the functional space. Species and communities that exhibit a high number of individuals and density of traits at the boundaries of the functional space are characterized by high functional divergence (Figs 1 and S4c in S1 File). In ecology, this indicates that the species within a community have differentiated niches, resulting in reduced competition for resources and conditions in the habitat [4]. In the case of species, high functional divergence suggests the presence of distinct populations within the same species that have adapted to exploit different resources and conditions in their habitat [5]. In the context of music, this concept can be understood as an artist releasing various types of songs, but with a small portion of those being popular–the songs situated at the extremes of the artist's functional space. For instance, Eminem has a diverse catalog spanning rap songs, energetic and danceable beats, and even highly acoustic songs (Fig 1C). However, the danceable and rap-oriented songs receive significantly more attention than the acoustic ones (Fig 1C).

## Effects of functional diversity on popularity

We evidenced that higher functional richness leads to a greater number of listens in Spotify, especially for artists that launched their career in recent years. The strength of this association is represented by an increase of 244,300,000 listens for each increase in functional richness, what translates to potential earnings of ~$1,067,591 for the artists or rights holders, assuming an average payment rate of $0.00437 per stream [41]. Consequently, emerging artists who consistently release distinct songs and diverse albums have the potential to reach a broader audience with varied musical preferences. However, it is worth noticing that, instead of targeting audiences with different "tastes", music labels and artists are targeting audiences from different social backgrounds. Personal preferences for specific types of music are constructed based on the cultural context of individuals [42, 43]. In this sense, "taste" is nothing but distinctions, that is, choices based on opposition to those chosen by other social groups [42]. The strategic decision for an artist to release different types of songs may be intentionally employed by music labels to ensure an artist's long-term relevance in the market by targeting different audiences and social groups [42–45].

However, this impact of functional richness on an artist's popularity depends on the year that an artist has launched their career. For instance, emerging artists (e.g., Taylor Swift) tend to attract more listeners and consequently generate greater profits on streaming platforms compared to established veterans (e.g., Paul McCartney) (see Figs 2A and S5 in S1 File). The landscape of music consumption has evolved significantly from the era of vinyl records [16, 46]. With the advent of the internet, digital music consumption through subscriptions (i.e., streaming services) has outpaced physical sales and even individual song purchases in digital platforms [28, 46]. As a result, the peak popularity achieved by artists who began their careers before the streaming era, such as in the 60s, 70s, and 80s, does not necessarily translate into success on streaming platforms during the 2010s decade. It is worth noticing here that richness, evenness and divergence and their relation to popularity tells us nothing about which specific characteristics make a song popular, but rather inform how rich and regular an artist is in their repertoire. To understand which characteristics makes a song popular, we have to mind their traits.

## Popularity of traits

Highly-energetic and danceable songs are inevitably more listened to than slow-paced songs, as they are fit for massive distribution at parties, concerts, and crowded events [17, 47]. However, this pattern does not necessarily mean that successful songs are happier. Interestingly

enough, songs that sound more positive are less listened to than their depressing counterparts. This evidence shows how cultural products of our society–i.e., music–accompany its current psychological state. Given that feelings of loneliness, social isolation, depression, and anxiety have been increasing since the past decade, especially during the COVID-19 pandemic [7, 17, 48].

Moreover, rock music seems to be less listened to than other genres of music, acknowledging its recent decrease in popularity. Rock has been a counterculture movement, often associated with dysphoric feelings of rebellion against what is popular and serving as a gateway for the anger of its listeners [49]. This still holds true today; however, young listeners between the ages of 18 to 35 –who constitute the majority among Spotify users [50]–are apparently more interested in pop-oriented and trap/hip-hop songs than rock [51]. A possible explanation for this pattern could lie in the prevalence of rock between the 1960s to the early 1990s, and consequently its popularity would be reflected in physical sales rather than streaming platforms.

Although patterns of valence and danceability remained unchanged when investigating which traits drive popularity within each musical genre, speechiness and energy showed contrasting patterns (S3 Fig in S1 File). We found that songs characterized by high speechiness are less popular in all musical genres but country. This finding indicates that country songs with elements of speech and spoken-word, which resemble rap songs, are more popular. Despite the differences between rap-oriented songs, which are often full of speech, and "honky-tonk" country songs, they share many similarities. Both rap and country music have their origins in Blues music, with pioneering African Americans musicians defining and revolutionizing the boundaries of these genres since the 1920s and 1970s, respectively [43, 52, 53]. Moreover, these two musical styles have been influencing each other for over 40 years, from Keef Cowboy of the Furious Five in the 1980s to Lil Nas X's 2019 hit "Old Town Road," despite resistance grounded on racism rhetoric of segregation and whitewashing of country music [52, 54]. Therefore, specific analyses tailored to different musical styles may reveal market tendencies unique to each genre, helping to target audiences more effectively.

It is worth noticing that we examined how different "traits"–characteristics of a song–influence popularity in the mainstream context. Consequently, our conclusions and speculations should be interpreted cautiously, as they reflect market patterns in mainstream media on a large scale, and, thus, are not cross-cultural. Studies at local scales that aim to apply the diversity metrics proposed here should first identify which traits are relevant to their specific context and how they are interpreted in that social setting. For example, the "valence" metric used in our study measures how positive a song sounds, which can vary greatly across cultures and social groups [47]. It is impossible to apply these metrics in a cross-cultural manner consistently, as interpretations change across space and over time. Danceability is another metric that should be interpreted differently across various audiences. "Bossa Nova", a musical style with Brazilian roots, has its own style of dancing [55], which suggest high danceability. However, this is not the case, as the gentle rhythm, jazzy harmonies, and smooth melodies of Bossa Nova are quite different from the high-tempo and upbeat Brazilian funk tracks played at major parties in Brazil. Therefore, our results are limited to the 2010s mainstream perspective of how songs affect people's emotions and should not be generalized to other social contexts.

Nevertheless, the traits and functional richness of artists are only part of what makes them more listened in the mainstream context. Massive marketing strategies by labels towards an artists' project could be vital at defining their success [56]. Otherwise, fandom engagement throughout social media could also play a vital role at making an artist more listened on streaming services [15, 19]. The success of a song may also rely on somewhat stochastic factors, such as a song becoming "viral" throughout social media in platforms such as TikTok [15]. Although some labels already plan on releasing songs specifically towards this end, it is up to

the users of social media to engage and turn a song into a viral phenomenon [15, 56]. There-fore, divulgation is likely one of the major drivers of a song, album or artist success on stream-ing services. Essentially, musical preferences of listeners are a construction of music sellers through statistical algorithms using user data [57], and thus depend on what is familiar to lis-teners of a specific social context [42, 43, 57, 58].

## Conclusions

In this study, we characterized the music of the most influential artists of the 2010s by applying concepts and methods of functional diversity to music research for the first time. In ecology, a functional trait refers to any morpho-physiological and behavioral characteristic of an organ-ism related to its fitness in a given habitat [6]. The term "trait" has also been incorporated into various other disciplines [2]. In nonscientific language, it refers to the characteristics of some-thing, often a person. The extensive literature and methods developed in functional ecology over the past three decades could be applied to other fields, including science and economics, where subjects of study can be represented by their traits.

While listeners may not be aware of music theory concepts, differences in sounds are intui-tive–this is how a fast-paced rap is distinguished from the evocative guitars of a rock song. Studying traits in music may reveal market trends over time and identify the traits that make a hit song, a valuable information for labels and music sellers [17]. However, the applications of functional diversity in music are context-dependent given that emotional perceptions of music and other arts–which will yield the traits of a song–depend on various cultural aspects [42, 47]. Despite these debates, the crossover between functional ecology and music can be useful in ecology classes, helping students understand complex concepts using familiar ideas–after all, music is universal [47]. After recognizing the power of the investigation of traits, we strongly believe that future possibilities for the application of functional diversity concepts into other fields of science and economics are endless and promising.

## Supporting information

**S1 File. Tables and figures supporting the main results.**
(PDF)

## Acknowledgments

We sincerely thank Maíra Cardoso, Fábio Alexandre, Vitor Schunemann, David Hoeinghaus and the attendees of the BES Macroecology meeting of 2023 at the University of Birmingham for the enlightening conversations about the scope of this paper. We also acknowledge the con-tributions of Cristian Dambros, Sâmia Leticia Reolon da Cruz and Felipe Cerezer for their valuable contributions on the writing of earlier versions of this paper.

## Author Contributions

**Conceptualization:** Lucas Colares, Ray Balieiro Lopes-Neto, Arianne Flexa de Castro.

**Data curation:** Lucas Colares.

**Formal analysis:** Lucas Colares.

**Investigation:** Lucas Colares, Ray Balieiro Lopes-Neto, Alexandre Sampaio de Siqueira, Camila Ferreira Leão.

**Methodology:** Lucas Colares, Bárbara Dunck.

**Project administration:** Lucas Colares.

**Resources:** Bárbara Dunck.

**Software:** Lucas Colares, Alexandre Sampaio de Siqueira, Camila Ferreira Leão.

**Supervision:** Bárbara Dunck.

**Visualization:** Lucas Colares.

**Writing – original draft:** Lucas Colares, Ray Balieiro Lopes-Neto, Alexandre Sampaio de Siqueira.

**Writing – review & editing:** Lucas Colares, Ray Balieiro Lopes-Neto, Alexandre Sampaio de Siqueira, Camila Ferreira Leão, Arianne Flexa de Castro, Bárbara Dunck.

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
