## [Decision Letter · Decision Letter 0]

19 Apr 2024

PONE-D-24-05432Functional diversity in human songPLOS ONE

Dear Dr. Colares,

Thank you for submitting your manuscript to PLOS ONE. After careful consideration, we feel that it has merit but does not fully meet PLOS ONE’s publication criteria as it currently stands. Therefore, we invite you to submit a revised version of the manuscript that addresses the points raised during the review process.

**I have now heard from two experts who are split in their final assessment of the paper. One accepts the basic premise of the paper and offers several wonderful suggestions and raises some questions that I am confident the authors can address. The other referee, a music scholar with work in AI, feels very negatively about the endeavor and writes "this kind of work is reckless and disrespectful of the work that many music scholars have done over decades." There is some agreement across the two reports. Both question whether the Spotify-based parameters are really commensurate with one another, which raises serious doubts about whether they can all be treated as dimensions in the way the authors do. Some of the parameters are actually potential abstractions of others; this doesn’t seem like a safe zone for considering them all as dimensions of the same kind. I am no expert in the field. But I do appreciate the tussle and its value. I am offering the authors a chance to revise their manuscript based on these reports. While I feel it would to too much to ask to entirely satisfy the critical referee, I would like the authors to make a serious effort - at the very least, be very clear on the limitations of what you are doing, and acknowledge the works of music scholars. **

We look forward to receiving your revised manuscript.

Kind regards,

Joydeep Bhattacharya

Academic Editor

PLOS ONE

Journal Requirements:

2. In your Methods section, please include additional information about your dataset and ensure that you have included a statement specifying whether the collection and analysis method complied with the terms and conditions for the source of the data.

3. Please remove your figures from within your manuscript file, leaving only the individual TIFF/EPS image files, uploaded separately. These will be automatically included in the reviewers’ PDF.

Reviewers' comments:

Reviewer's Responses to Questions

**Comments to the Author**

1. Is the manuscript technically sound, and do the data support the conclusions?

Reviewer #1: Partly

Reviewer #2: Partly

2. Has the statistical analysis been performed appropriately and rigorously? 

Reviewer #1: I Don't Know

Reviewer #2: I Don't Know

3. Have the authors made all data underlying the findings in their manuscript fully available?

Reviewer #1: Yes

Reviewer #2: Yes

4. Is the manuscript presented in an intelligible fashion and written in standard English?

Reviewer #1: Yes

Reviewer #2: Yes

5. Review Comments to the Author

**Reviewer #1:** The study presented by the authors examines the role that functional diversity, or the distribution of characteristics in a community, plays in the popularity of songs across time. Examining a corpus of 13000 songs from the top 100 artists throughout the 2010s, the authors argue that at least one measure of functional diversity, functional richness, correlates with increased the popularity of music across time. In addition to this, the authors examine other characteristics of these songs along fourteen dimensions, thirteen of which are extracted directly from Spotify’s API, finding that highly danceable songs are preferred across time.

The primary contribution of the current study is in its use of measures of functional diversity to characterize the bodies of work of the Top 100 artists and attach them to a real-world metric, in this case, popularity in terms of the number of listens. While the results themselves seem promising, especially in terms of providing us with the ability to describe within-artist corpus distributions in a unique way, I believe the study could use more work in describing what these measures, in a more precise, formal way, are and how they relate to one another. Particularly, the section beginning on page 8, Functional indexes, needs to be extended to make the study clearer. For example, what exactly is the probabilistic hypervolume method and the probability density used therein?

More critically, because the functional indexes are calculated within each artist, I would like to ask how useful is it to compare some of these functional indexes across artists? Is it the case that artists with the exact same functional richness, for example, are doing the same thing in trait space. In other words, are these dimensions stable across artists? Examining Figure 1c and 1l give the appearance that while Eminem and Calvin Harris are both divergent within their own domains, to make some functional claim about the importance of divergence on listeners based on these two might be a mistake because they diverge in different areas of state space, so some nuance may be necessary with regards to the authors’ null findings rather than the assumption that these traits should be endogenous drivers of popularity on their own.

Several other methodological questions follow:

- How much is the volume measure of the hypervolumes driven by outliers (lls 169-171), as in the case of functional richness?

- How much do richness, evenness, and divergence interplay with one another? I can see a world where an artist is extremely even specifically because they are neither rich nor divergent. Is there some straightforward means for assessing their interaction?

- To what extent can the results in Figure 3 be broken down by each genre? I am fine with the global comparison lumping all genres together, but it’s not clear to me that people are listening to the same genres of music for the same reasons and so the utility of the lumping here is not straightforward.

I have a couple of line comments with regards to the study below:

Abstract, lls (40-41): “We captured how patterns in human song reflects the social state of human societies through time.” - I am not sure about this broad of a claim. To what extent does the study discuss the social state of human societies through time?

Figure 1 caption: Why are the subplots labeled “adgjmpsvyB”, “behknqtwzC”, etc?

**Reviewer #2**: Dear Authors,

Thank you for producing this work. I have appreciated the opportunity to get to know this project.

The paper takes on an ambitious and important question: what drives popularity for artists and tracks on contemporary streaming platforms? While I am sympathetic to many of the goals of this paper, I find that it ignores large bodies of relevant research and rests upon assumptions that no expert in the study of music can agree with. Thus, I must unfortunately recommend that this paper be rejected, particularly as it is likely to perpetuate many misunderstandings about the study of music that currently circulate in the natural and applied sciences.

One of the many major flaws of this paper is that it assumes that principles of analysis and empirical findings from the natural and biological sciences are directly applicable to the social sciences and the study of expressive culture (i.e., the humanities). To be fair, I will not deny that culture and social behavior are not at least partially shaped by biology. But the very fact that I am writing to you in English and not Portuguese or Wolof ought to already be evidence enough that there is much about human behavior that at the very least can’t be explained by biology alone or at least should give us a lot of pause in directly applying theories from biology wholesale to the study of music, which is currently an alarming current scholarly trend. The justification for this kind of thinking is very thinly given in this paper. For example, on pp. 20-21 of the MS the authors suggest that the fact that the term “trait” is used both by biologists and non-biologists suggests that the term’s meaning out to be more or less similar across domains. This is utter nonsense. Overall, the whole analysis rests upon a rather bizarre metaphor in which songs are compared to species. This is fascinating and imaginative, but we must immediately recognize that these things are not the same, and that any responsible comparison between them requires us to recognize their differences first. Songs are created by people; only a fraction of species may have ever had this attribute, if any. To really take that metaphor seriously borders on the theological-scientific theory of “intelligent design.”

To make this point more clearly, I’ll turn to the second (and perhaps most significant) error in this analysis. The authors take many phenomena, such as an artist’s number of streams on Spotify, as “natural” phenomena when they are absolutely anything but. I hate to be so blunt, but I cannot say less than this: the authors show exceptional disregard and near total ignorance of most social scientific and humanistic scholarship on music. This alone is not an error, but by ignoring this work, the authors ultimately misinterpret much of what they attempt to analysis. They fail to sufficiently or substantively consider the fact that the popularity — even the commercial viability and existence — of particular artists is not a produce of the general population’s “natural” taste for their work, but rather the decision of executives, producers, and marketing professionals in global music industries. “Taste” itself is not a natural phenomenon; it is often a product of our presence in various social groupings. For further clarification on this point, I urge the authors to consult sociologist Pierre Bourdieu’s monograph Distinction (Harvard UP 1984). On the point that the popularity and existence of certain artists is a corporate, and not public/natural, decision, see Segregating Sound by Karl H. Miller (Duke UP 2010) among countless others. I really recommend Miller above all else because it is a careful study of the way that producers in the early 20th century American music industry systematically suppressed what the authors might call the “functional divergence” of artists they chose to record by dictating what they could and could not record. The decision of which artist to promote (or even record and produce for that matter) is not one of biology, but of culture. I urge the authors to take the distinction between biology and culture seriously. I also remind them that the failure to distinguish between these two things has at least one name: “biological determinism” at its most benign and “scientific racism” at its most toxic, this being the belief that one’s “race” (which most social scientists and even biologists recognize to be a complete fiction for our species) determines one’s behavior and life outcomes.

Another major flaw of this paper is the extraordinarily naive interpretation the authors make of the metadata that Spotify attaches to tracks in its library. I am referring to the characteristics such as key, mode, time signature, duration, acousticness, etc. that the authors “extracted” using the Spotify API (extraction of these features is now trivial; the authors present this as if it were their innovation). Like many other researchers stumbling upon this metadata, the authors presume that it can be applied to a pure scientific question even though the metadata has been cooked up for the very clearly applied scientific purpose of music recommendation. They also assume that the features Spotify has concocted for these purposes are actually relevant. Most listeners cannot discern what key a song is in, though they can often distinguish between tonalities (i.e., major, minor, diminished, etc.) even if they can’t explicitly describe the difference they are hearing. They also pay no attention to the fact that the 13 parameters they extract from Spotify data are not at all alike with one another; some are lower level features like key or mode, and others are complicated abstractions of these (such as “danceability,” “energy,” or “valence”). It is excusable for a non-scholarly party encountering these parameters to take them at face value, but here we must apply a higher standard. “Valence” is a particularly suspicious attribute: please tell me how we are supposed to accurately and objectively devise a strategy for computing the “positivity” of millions of tracks in manner which is transculturally and ahistorically valid. There is no way to do this. Similarly, “danceability” assumes that all human beings have the same proclivity to dance when exposed to particular musical stimuli. For a paper that so frequently mentions “diversity,” this is the negation of any true respect for this concept since many musical forms considered “danceable” in one sociocultural milieu will be considered “undanceable” in another. For this point, I refer the authors to K. Goldschmitt’s 2011 article in the Luso-Brazilian Review, which offers a history of bossa nova as not a genre for listening, but one for dancing. In any event, the authors claim that a set of traits they choose from Spotify’s metadata “[paints] a full picture of every aspect of a song” (p. 7). This is patently false. Even if they were to revise this one sentence, the rest of the analysis is based in this logic and what’s needed is a fundamentally different framework of analysis; as noted above, this framework exists in spades in the scholarly study of music and yet the authors largely disavow its existence.

This leads us to yet another problem: the authors assume that Spotify listens are actually a product of what the music sounds like. Over and over again, scholars of music have established that this is not the case. To offer a more proximate reference, please read cultural anthropologist Nick Seaver’s recent book Computing Taste (UChicago Press, 2022). This book is literally an ethnography of the engineers that have designed platforms like Spotify and related algorithmic music recommendation tools. What’s established there and elsewhere is the fact that listeners feel that they are drawn to particular artists not because of their sound intrinsically but because that sound is one that is familiar to them. If the domain of cultural anthropology is not particularly credible to these authors, then perhaps psychology might be. If that’s the case, a quick perusal of Robert Zajonc’s classic (1968) paper on the exposure effect might help, in which he argues that we are more drawn to things to which we have been previously exposed. The basic conceit of that paper has been established in metastudies over and over since its first publication more than half a century ago.

I also find that the authors have arbitrarily chosen to exclude some of the data that might contradict their basic hypotheses. As they note on p. 7, they have “considered only musical genres that appeared in at least 5% of all 10,444 songs.” I recognize that functional diversity and “diversity” as a term in current social and political debates are not the same. This seems like an analytical choice that requires more explanation at the very least and signals an arbitrary and self-interested analytical exclusion. At the very least, it seems strange not to test that 5% of the data for diversity, especially when it seems to retain this feature.

I would like the authors to also consider the layout and user experience of Spotify as a factor that drives popularity. The authors imagine a hermetically sealed world in this paper in which users think of a particular artist and their sounds materialize at will. The reality is much less immediate; the interface of Spotify directs users through the computational arts of persuasion to particular artists. Their “popularity” is manufactured through these design choices. The authors write as if all this is irrelevant.

In light of the above, I’ll note that the authors do mention marketing in a handful of locations. This is insufficient to undo the damage of their basic assumptions.

Despite its flaws, I still find that this paper offers some really surprising and potentially insightful observations. Taking everything noted above into consideration, I found the observation about a correlation between “functional diversity” and streaming popularity quite surprising. I also found the use of hypervolumes interesting, though this is a natural and incremental extension from the vector-based analytic methods often used in music information retrieval. I had not realized that XXXtentacion was so varied (allegedly) in their stylistic tendencies. Then again, there is a reason why this might not be apparent since the kind of stylistic variation he produces is ultimately still within a relatively small range of music-stylistic possibilities.

In closing, I sympathize with this paper’s ambitions in understanding the driving factors of popular music. Ultimately, the paper has many flaws, most which hang from its disattention to the basic differences between biology and culture. Regarding the publication of this paper, I again recommend rejection even despite the relative creativity of this analysis. This is mainly due to the flaws of the paper but also due to the fact that we are currently witnessing a great expansion of scholarship ostensibly on music that takes this approach with many of the same flaws; work by Samuel Mehr, who is referenced here as a touchstone, is a classic example. Most music scholars are openly skeptical of this work; much of what we have studied immediately illustrates that these “big data” approaches are nonstarters. In this context, this work is not likely to be read by music scholars. So then this leaves as the putative audience other scholars interested in biological metaphors and with a relatively basic understanding of music. To publish this work is to further misunderstandings among those who already have a weak comprehension of the topic and who by nature, it seems, regard those who have dedicated their time to the subject of music as being largely irrelevant when it comes to the study of this topic.

6. PLOS authors have the option to publish the peer review history of their article (what does this mean?). If published, this will include your full peer review and any attached files.

Reviewer #1: No

Reviewer #2: No

---

## [Author Response · Author response to Decision Letter 0]

26 Jun 2024

Response letter to the editor and reviewers of PLOS ONE

Editor’s comments (Joydeep Bhattacharya):

Editor comment: I have now heard from two experts who are split in their final assessment of the paper. One accepts the basic premise of the paper and offers several wonderful suggestions and raises some questions that I am confident the authors can address. The other referee, a music scholar with work in AI, feels very negatively about the endeavor and writes "this kind of work is reckless and disrespectful of the work that many music scholars have done over decades." There is some agreement across the two reports. Both question whether the Spotify-based parameters are really commensurate with one another, which raises serious doubts about whether they can all be treated as dimensions in the way the authors do. Some of the parameters are actually potential abstractions of others; this doesn’t seem like a safe zone for considering them all as dimensions of the same kind. I am no expert in the field. But I do appreciate the tussle and its value. I am offering the authors a chance to revise their manuscript based on these reports. While I feel it would to too much to ask to entirely satisfy the critical referee, I would like the authors to make a serious effort - at the very least, be very clear on the limitations of what you are doing, and acknowledge the works of music scholars. 

Author response: Dear Joydeep Bhattacharya, thank you for handling our paper on behalf of PLOS One. We appreciate your time and dedication. We are glad to inform you that we have considered all comments from both reviewers during the revision of this paper. We clarified how the traits and indexes of functional diversity should be interpreted in the context of our data (lines 105-126, 184-191, and 314-353), which was one of the main concerns of reviewer #1. We also tried to address the suggestions of reviewer #2 to the best of our ability. In this new version of the manuscript, we have acknowledged the works of music scholars that were suggested, in addition to those we already cite throughout the text. Finally, we modified our discussion section to acknowledge the cultural limitations of applying such functional metrics to subject areas in which traits are related to the emotional perceptions and social backgrounds of an audience (lines 361-368 and 413-427). We hope that these alterations satisfy your concerns regarding this paper, as well as the suggestions of both reviewers. Thank you again for considering this paper for publication in PLOS One, our top-ranking journal.

Editor comment: In your Methods section, please include additional information about your dataset and ensure that you have included a statement specifying whether the collection and analysis method complied with the terms and conditions for the source of the data.

Author response: In this new version of the manuscript, we have provided further details on our trait dataset (lines 105-126) and added a statement that all our data collection and analysis comply with the terms and conditions of the Spotify and Last.fm application programming interfaces (lines 149-150).

Reviewer #1

Reviewer comment: The study presented by the authors examines the role that functional diversity, or the distribution of characteristics in a community, plays in the popularity of songs across time. Examining a corpus of 13000 songs from the top 100 artists throughout the 2010s, the authors argue that at least one measure of functional diversity, functional richness, correlates with increased the popularity of music across time. In addition to this, the authors examine other characteristics of these songs along fourteen dimensions, thirteen of which are extracted directly from Spotify’s API, finding that highly danceable songs are preferred across time. The primary contribution of the current study is in its use of measures of functional diversity to characterize the bodies of work of the Top 100 artists and attach them to a real-world metric, in this case, popularity in terms of the number of listens. While the results themselves seem promising, especially in terms of providing us with the ability to describe within-artist corpus distributions in a unique way, I believe the study could use more work in describing what these measures, in a more precise, formal way, are and how they relate to one another. Particularly, the section beginning on page 8, Functional indexes, needs to be extended to make the study clearer. For example, what exactly is the probabilistic hypervolume method and the probability density used therein? More critically, because the functional indexes are calculated within each artist, I would like to ask how useful is it to compare some of these functional indexes across artists? Is it the case that artists with the exact same functional richness, for example, are doing the same thing in trait space. In other words, are these dimensions stable across artists? Examining Figure 1c and 1l give the appearance that while Eminem and Calvin Harris are both divergent within their own domains, to make some functional claim about the importance of divergence on listeners based on these two might be a mistake because they diverge in different areas of state space, so some nuance may be necessary with regards to the authors’ null findings rather than the assumption that these traits should be endogenous drivers of popularity on their own.

Author response: We appreciate your constructive and in-depth suggestions on our paper and are pleased to inform you that we were able to apply every suggestion you made, which helped us improve our paper in this new version. We described the functional traits in more detail (lines 105-126) and explained the functional indexes we calculated (lines 184-191). Furthermore, we provided a more thorough explanation of the probabilistic hypervolume method (lines 152-173), hoping it meets your expectations. We included additional disclaimers on how to interpret the functional diversity metrics (i.e., richness, evenness, and divergence) in the methods (lines 184-191) and discussion sections (lines 314-353 and 378-381), explaining that different artists may have similar values of richness, for example, but this does not mean they have a similar body of work. Rather, it indicates a similar configuration of their functional space within their own domains. We provide further details on your minor suggestions below.

Reviewer comment: How much is the volume measure of the hypervolumes driven by outliers (lls 169-171), as in the case of functional richness?

Author response: We controlled for outliers in the data by adopting a density threshold of 0.99 instead of 1 (lines 153-157). This approach restricts the estimations to 99% of the data, excluding the 1% that might be outliers, as recommended in Carmona, C. P., de Bello, F., Mason, N. W., & Lepš, J. (2019). Trait Probability Density (TPD): measuring functional diversity across scales based on trait probability density with R. Ecology. doi:10.1002/ecy.2876.

Reviewer comment: How much do richness, evenness, and divergence interplay with one another? I can see a world where an artist is extremely even specifically because they are neither rich nor divergent. Is there some straightforward means for assessing their interaction?

Author response: Indeed, this is a possibility. In this new version of the manuscript, we consider the interactions between richness, evenness, and divergence in our linear models. We describe this in the methods section (lines 193-209). However, our results remain unchanged (lines 268-275; S7 Table), suggesting that the three metrics do not interact with one another

Reviewer comment: To what extent can the results in Figure 3 be broken down by each genre? I am fine with the global comparison lumping all genres together, but it’s not clear to me that people are listening to the same genres of music for the same reasons and so the utility of the lumping here is not straightforward.

Author response: We subdivided Figure 3 to represent the influence of traits on the number of streams within each musical genre. However, we were unable to provide results on which traits drive the popularity of R&B, Pop/Rock, and Trap artists due to insufficient replicates for our linear models. Nonetheless, you can still check the traits that drive popularity in Country, Pop, Rock, and Rap at the artist level, and in all seven major musical genres at the album and track levels. As shown in S3 Figure, the overall pattern of trait influence on track popularity remains consistent across all musical genres. For example, the association between valence and popularity, discussed in the main text, is negative for all genres except Country and Trap (in which valence was not selected as an important variable to explain track popularity). The same pattern is observed for speechiness, liveness, tempo, instrumentalness, acousticness, and danceability. The only traits that show contrasting patterns between musical genres are energy, which has a positive influence on track popularity across all genres except Trap, and duration, which has a positive influence on track popularity across all genres except Rock and Pop/Rock. These contrasting responses are briefly explored in the results and discussion sections (lines 296-303 and 399-412, respectively). Since most patterns follow the same direction of influence across genres, we chose to maintain the global comparison plot in the main text while presenting the panels by musical genre in S3 Figure.

Reviewer comment: Abstract, lls (40-41): “We captured how patterns in human song reflects the social state of human societies through time.” - I am not sure about this broad of a claim. To what extent does the study discuss the social state of human societies through time?

Author response: We changed the sentence to “We captured how patterns in human song might reflect the social state of human societies in recent years”. We discuss these matters speculatively, but grounded in previously cited literature, in lines 383-398 of the discussion.

Reviewer comment: Figure 1 caption: Why are the subplots labeled “adgjmpsvyB”, “behknqtwzC”, etc?

Author response: We used letters to refer to the specific panels that show richness, evenness, and divergence. We understand that the large number of panels can be confusing, so we rewrote the legend of Figure 1 (lines 260-266).

Reviewer #2

Reviewer comment: Dear Authors, thank you for producing this work. I have appreciated the opportunity to get to know this project. The paper takes on an ambitious and important question: what drives popularity for artists and tracks on contemporary streaming platforms? While I am sympathetic to many of the goals of this paper, I find that it ignores large bodies of relevant research and rests upon assumptions that no expert in the study of music can agree with. Thus, I must unfortunately recommend that this paper be rejected, particularly as it is likely to perpetuate many misunderstandings about the study of music that currently circulate in the natural and applied sciences.

Author response: Dear Reviewer, thank you for your responses to our paper. We acknowledge that our framework has its limitations and do not hide those in this new version (see lines 413-427). We appreciate all the constructive suggestions you made for the improvement of our paper and are glad to inform you that we tried to apply those in the best way we could find. In this new version of the manuscript, you will find a discussion more focused on the cultural limitations of our assessment and how our results may relate to the social backgrounds of the listeners (lines 361-368). We believe that a paper describing market patterns at a large scale, such as ours, should not be invalidated as these results are directly applicable to mainstream media. Like any study that chooses to address large-scale patterns, our study has limitations regarding the spatial resolution of our patterns. We do not intend for our results to be directly applicable to local contexts at small spatial scales, as the emotional perceptions of listeners will be constrained by social and cultural contexts, as you acknowledge several times in your critical review. We are glad to inform you that we mention these limitations in this new version of the manuscript.

Reviewer comment: One of the many major flaws of this paper is that it assumes that principles of analysis and empirical findings from the natural and biological sciences are directly applicable to the social sciences and the study of expressive culture (i.e., the humanities). To be fair, I will not deny that culture and social behavior are not at least partially shaped by biology. But the very fact that I am writing to you in English and not Portuguese or Wolof ought to already be evidence enough that there is much about human behavior that at the very least can’t be explained by biology alone or at least should give us a lot of pause in directly applying theories from biology wholesale to the study of music, which is currently an alarming current scholarly trend. The justification for this kind of thinking is very thinly given in this paper. For example, on pp. 20-21 of the MS the authors suggest that the fact that the term “trait” is used both by biologists and non-biologists suggests that the term’s meaning out to be more or less similar across domains. This is utter nonsense. Overall, the whole analysis rests upon a rather bizarre metaphor in which songs are compared to species. This is fascinating and imaginative, but we must immediately recognize that these things are not the same, and that any responsible comparison between them requires us to recognize their differences first. Songs are created by people; only a fraction of species may have ever had this attribute, if any. To really take that metaphor seriously borders on the theological-scientific theory of “intelligent design.”

Author response: First and foremost, and as we make clear in the main text, we do not intend to directly use concepts of biological sciences to explain patterns in social sciences. However, we recognize the close relationship between these fields, and anyone who believes otherwise would be greatly mistaken. Our intentions are to demonstrate how specific methods and mathematics tailored in a biological context can also be applied to other disciplines, which is entirely plausible. Nowadays, many fields apply analyses and methods developed by mathematics and statistics to both social and natural sciences to differentiate empirical findings from null hypotheses. We follow this same principle but in the opposite direction, using biological examples to illustrate the applicability of such methods. It's important to note that we did not assert in our paper that songs are directly comparable to species. Instead, we aim to show how two distinct contexts can be interpreted using mathematical indexes to measure the diversity of traits, which is a familiar concept across disciplines – referring to the characteristics or qualities of something, from species to genes, songs, and people.

Reviewer comment: To make this point more clearly, I’ll turn to the second (and perhaps most significant) error in this analysis. The authors take many phenomena, such as an artist’s number of streams on Spotify, as “natural” phenomena when they are absolutely anything but. I hate to be so blunt, but I cannot say less than this: the authors show exceptional disregard and near total ignorance of most social scientific and humanistic scholarship on music. This alone is not an error, but by ignoring this work, the authors ultimately misinterpret much of what they attempt to analysis. They fail to sufficiently or substantively consider the fact that the popularity — even the commercial viability and existence — of particular artists is not a produce of the general population’s “natural” taste for their work, but rather the decision of executives, producers, and marketing professionals in global music industries. “Taste” itself is not a natural phenomenon; it is often a product of our presence in various social groupings. For further clarification on this point, I urge the authors to consult sociologist

---

## [Editor Report · Decision Letter 1]

28 Jun 2024

Functional diversity in human song

PONE-D-24-05432R1

Dear Dr. Colares,

We’re pleased to inform you that your manuscript has been judged scientifically suitable for publication and will be formally accepted for publication once it meets all outstanding technical requirements.

Kind regards,

Joydeep Bhattacharya

Academic Editor

PLOS ONE
---

## [Editor Report · Acceptance letter]

3 Jul 2024

PONE-D-24-05432R1 

PLOS ONE

Dear Dr. Colares, 

I'm pleased to inform you that your manuscript has been deemed suitable for publication in PLOS ONE. Congratulations! Your manuscript is now being handed over to our production team.

Kind regards, 

on behalf of

Dr. Joydeep Bhattacharya 

Academic Editor

PLOS ONE